# Estimating Running Ground Reaction Forces from Plantar Pressure during Graded Running

**DOI:** 10.3390/s22093338

**Published:** 2022-04-27

**Authors:** Eric C. Honert, Fabian Hoitz, Sam Blades, Sandro R. Nigg, Benno M. Nigg

**Affiliations:** 1Human Performance Laboratory, Department of Kinesiology, University of Calgary, Calgary, AB T2N 1N4, Canada; fabian.hoitz@ucalgary.ca (F.H.); sandro.nigg@ucalgary.ca (S.R.N.); nigg@ucalgary.ca (B.M.N.); 2Kinetyx Sciences Inc., Calgary, AB T2G 1M8, Canada; sam.blades@kinetyx.tech

**Keywords:** wearable technology, machine learning, recurrent neural network, statistical parametric mapping, gait analysis

## Abstract

Ground reaction forces (GRFs) describe how runners interact with their surroundings and provide the basis for computing inverse dynamics. Wearable technology can predict time−continuous GRFs during walking and running; however, the majority of GRF predictions examine level ground locomotion. The purpose of this manuscript was to predict vertical and anterior–posterior GRFs across different speeds and slopes. Eighteen recreationally active subjects ran on an instrumented treadmill while we collected GRFs and plantar pressure. Subjects ran on level ground at 2.6, 3.0, 3.4, and 3.8 m/s, six degrees inclined at 2.6, 2.8, and 3.0 m/s, and six degrees declined at 2.6, 2.8, 3.0, and 3.4 m/s. We estimated GRFs using a set of linear models and a recurrent neural network, which used speed, slope, and plantar pressure as inputs. We also tested eliminating speed and slope as inputs. The recurrent neural network outperformed the linear model across all conditions, especially with the prediction of anterior–posterior GRFs. Eliminating speed and slope as model inputs had little effect on performance. We also demonstrate that subject−specific model training can reduce errors from 8% to 3%. With such low errors, researchers can use these wearable−based GRFs to understand running performance or injuries in real−world settings.

## 1. Introduction

Ground reaction forces (GRFs) describe how runners interact with their surroundings and provide the basis for computing inverse dynamics for biomechanical studies. In the laboratory, these GRFs are measured via force plates imbedded in the ground. These forces can then be used to estimate valuable metrics, such as tibia bone loads, that can help in understanding overuse injuries [1] or lower limb joint powers related to understanding the performance benefits of shoes [2]. Yet, there is an increasing need to compute and track such metrics *outside of the laboratory* to better understand overuse injuries and/or real−time performance. Wearable technology can be used to quantify biomechanical metrics [3,4] and/or provide biomechanical feedback to athletes outside of a laboratory setting (for a review see [5]). Specifically, wearables can be utilized to predict time−continuous GRFs during walking [6,7,8,9] and running [10,11,12]. However, the majority of time−continuous GRF prediction research has been focused on level ground locomotion [4,10,11,13,14,15,16,17,18,19,20]. Yet, during outdoor running, athletes encounter a wide variety of different speeds [21] and slopes [22,23], which influences lower limb kinetics [24]. As such, if wearables are to be effectively used outside of the laboratory to assess injury and performance, they need to be able to estimate GRFs across different speeds and slopes.

This gap is starting to be filled as recently wearables have been used to compute vertical GRFs at different speeds and slopes [12]. The vertical GRF component can provide valuable feedback for athletes outside the laboratory; however, if wearable−based GRFs are to be used to compute additional kinetic metrics, multiple components of the GRF should be estimated. For example, Center−of−Mass (COM) power [25], which is computed from GRFs and running speed and slope, can provide insights into the contributions of the GRF components to the total COM power that is performed by the lower limbs (Figure 1). Note that the COM power is essentially the summation of the ankle, knee, and hip powers [26]. The vertical component of the COM power, exclusively computed from the vertical GRF, only provides a portion of the total power performed by the lower limbs (Figure 1). Including the anterior–posterior (A/P) GRF into the COM power demonstrates that the sagittal plane GRFs have the ability to account for the vast majority of the power provided by the lower limbs. As such, wearable technology that predicts GRFs should estimate both vertical and A/P GRFs if the aim is to compute further biomechanical metrics outside of the laboratory.

Another overlooked aspect of many GRF predictions is understanding where in the step misestimates occur. Despite the time−continuous nature of the GRF, it is common for model estimates to be validated using summary metrics (i.e., root mean squared error, correlation, [6,7]) or discrete metrics (i.e., peak GRF, [4,14,16,20]). Furthermore, many GRF predictions have focused on cohort level outcomes and it is currently unknown how well GRFs can be predicted for individual subjects. Statistical parametric mapping (SPM) is sometimes used in biomechanical analyses in order to evaluate the differences between time−continuous curves [27] and can be used to evaluate both cohort level and individual subject differences [28]. This methodology can be powerful on its own; however, it only provides an estimate of when significant differences occur. Additional time−continuous errors are needed to supplement the SPM to also understand the magnitude of difference throughout the steps. Understanding the accuracy of time−continuous GRF predictions would provide confidence for researchers who want to estimate GRFs outside of the laboratory for athlete monitoring or for performing basic science research.

Thus, the purpose of this manuscript was to predict sagittal plane GRFs across different speeds and slopes, and to understand the time−continuous accuracy of the predictions on both a cohort and individual subject basis.

The main contributions of this work are:The first running study to predict multiple ground reaction force components during running for different speeds and slopesWe introduce a new combination of tools to understand the performance of time−continuous model predictions during gaitGRF predictions with plantar pressure do not need a priori knowledge of the speed or slopeSubject−specific training can enhance GRF predictions, such that these predictions could be confidently used outside of the laboratory

## 2. Materials and Methods

### 2.1. Participants and Protocol

Eighteen recreationally active subjects (9 males, 9 females; age: 28 ± 5 years; Height: 1.73 ± 0.11 m; mass: 65.9 ± 9.3 kg) provided written informed consent and participated in this study. The protocol was approved by the University of Calgary’s Conjoint Health Research Ethics Board (REB20−1734). A similar study size was previously utilized to examine the performance of machine learning algorithms predicting GRFs [12]. Participants ran on an instrumented treadmill while we collected GRFs (2400 Hz, Bertec, Columbus, USA) and Pedar plantar pressure (100 Hz, Novel, Munich, DEU) from both the left and right feet. Subjects ran on level ground at 2.6, 3.0, 3.4, and 3.8 m/s, six degrees inclined at 2.6, 2.8, and 3.0 m/s, and six degrees declined at 2.6, 2.8, 3.0, and 3.4 m/s. Data were collected for 75 s. Subjects were not told to conform to a certain foot−striking strategy; as such, most participants utilized a rearfoot striking strategy while others exhibited a forefoot striking strategy while running on level ground. At the beginning of the trial, subjects were asked to stand quietly on the treadmill belt and then perform three jumps. These jumps synchronized the two independent data collection modalities so that similar steps could then be extracted post−hoc. After the jumps were performed, the subjects were brought up to the desired running speed. To ensure only steady−state running was examined, data were analyzed from 25 s onward in the trial. In several instances, subjects were not able to maintain the desired running speed for the entire trial, particularly the uphill running trials. In these cases, subjects were encouraged to run as long as possible at the desired speed until they indicated that they could not maintain the speed. At this point, the treadmill was stopped and data collection for the trial was terminated. 

### 2.2. Data Processing

Ground reaction forces were first rotated such that the vertical component of the GRF was parallel to gravity in the sloped running conditions. The GRFs were then bi−pass filtered using a 50 Hz, third order, low−pass Butterworth filter. Next, GRF and plantar pressure data were aligned based on the three data synchronizing jumps through a cross−correlation. Both GRF and plantar pressure data were then segmented into similar steps for both the left and right foot, based on their respective signals. The plantar pressure data were then summed in five different regions to provide five time−continuous signals (Figure 2). All data, for each step, were then interpolated to 101 points as inputs to the different GRF prediction algorithms. All data processing occurred in MATLAB (Mathworks, Nantick, USA).

### 2.3. Model Development: Linear Model

We first estimated GRFs using linear regression models to provide a baseline comparison for the regression style machine learning model that is outlined below.

We created a set of least square linear models to predict the vertical (Figure 3, GRF_vert_) and anterior/posterior GRFs (Figure 3, GRF_A/P_) from five different plantar pressure regions (PR_1_−PR_5_, Figure 2 and Figure 3). This linear regression accounted for speed and slope by solving for the regression constants A_1_−A_5_ and B_1_−B_5_ for each different condition. This effectively enabled speed and slope to be independent predictors. The summed pressures from the five regions of the Pedar insoles (PR_1_−PR_5_) were normalized by subject body mass. This analysis was performed with these five different pressure regions in order to make the results presented here applicable to other plantar pressure sensors that may not have as fine a spatial resolution.

We next created a set of linear models to understand the effect of different running speeds and running slopes on this model type. In this case, we created two linear models to predict the vertical and anterior–posterior GRFs, respectively, to predict all running conditions. In this case, the regression constants A_1_–A_5_ and B_1_–B_5_ (Figure 3) were each solved for once. Similar to the first set of linear models, only the mass normalized insole pressures from the five different pressure regions were utilized to train the linear model.

### 2.4. Model Development: Recurrent Neural Network

We then designed a single regression recurrent neural network (RNN) to predict vertical and anterior–posterior ground reaction forces (Figure 3). Recurrent neural networks have been used previously to estimate vertical GRFs [12]. Using such an algorithm allows for a more flexible solution than a linear model, as the network is not restricted to one certain type of mathematical operation. The model consisted of 11 individual layers: a sequence input layer that processed eight predictor variables per time point (five pressure regions, mass, speed, and slope), a bidirectional long short−term memory layer (BiLSTM) with 400 nodes, a 30% dropout layer, a BiLSTM with 200 nodes, a 20% dropout layer, two fully connected layers (FC) with 300 and 150 nodes, and a final FC with 2 nodes that passed activations to a regression output layer. The BiLSTM and FC layers utilized hyperbolic tangent activations (Figure 3). For more information regarding network training options and hyperparameters see the supplementary data. Running speed and slope were based on the set treadmill speed and slope. Network training was performed via the adaptive moment estimation optimizer. Prior to any training, all input sequences were shuffled to avoid order effects and input sequences were normalized using Z−Score normalization. We designed and tested a similar network without the speed and slope predictors in order to determine the effects of these predictors.

### 2.5. Validation

For all models, validation was performed through a leave−one−subject−out cross validation design [29]. This validation method trains the model on 17 of the 18 subjects, then that trained model is used to predict the ground reaction forces for the one testing subject that was not in the training set. This process is then repeated for each subject. For each testing subject, we report the root mean squared error (RMSE) and Pearson’s correlation coefficient (*r*) to assess the model performance, similar to other models that predict GRFs (e.g., [6,7]). Left and right foot GRFs were evaluated together. 

As both the RMSE and correlation coefficient are summary metrics for time−continuous curves, we then performed statistical parametric mapping (SPM, [27]) and estimated a time−continuous percent error in order to understand *where* systematic differences between model predictions and the ground truth occurred. We first performed a SPM analysis to determine differences between subject−averaged curves. For each condition, we performed two−tailed paired SPM *t*−tests using spm1d (version M.0.4.7, www.spm1d.org, accessed on 18 March 2020) comparing the GRFs from the instrumented treadmill to each of the models. The SPM paired *t*−tests only provide information on whether there is a significant difference. We also calculated the absolute error between the models and the GRF throughout the stance phase to understand the difference magnitude. This error was normalized, based on range of the GRF for a particular step. It is important to note that a high percent error may not correspond to a significant difference and that a low percent error may correspond to a significant difference. This is due to the nature of the statistical test. For example, if a paired t−test is performed on two conditions and one condition has a slightly higher mean because all (or almost all) participants were affected by the condition, there would be a significant difference no matter the magnitude of the difference. The significance only corresponds to whether there is a systematic difference. On the other hand, if one condition had a large, but random, effect on the participants, it could result in no statistical difference. Yet, such an effect could result in a large absolute error. 

Next, we performed a similar analysis for each individual subject. For each subject and condition, a SPM paired t−test was performed between the model predictions and the GRFs measured from the instrumented treadmill. In this case, 18 randomly selected steps were used to perform the SPM in order to keep the statistical power similar to the SPM tests outlined in the previous paragraph. For these 18 steps, the absolute percent error was calculated. As it would be overwhelming to present results from each subject, condition and model, three subjects for the vertical GRF and three subjects for the A/P GRF are shown to illustrate low, average, and high percent errors for a single condition for the recurrent neural network. 

## 3. Results

All subjects across all conditions contributed to the training and testing datasets. In total, 24,882 steps were analyzed. Model performances with changes in training dataset size can be seen in Appendix A (Figure A1).

### 3.1. Model Ability to Predict Average Ground Reaction Forces

Across all speeds, slopes, and GRF components, the recurrent neural network had lower RMSE and equal or higher correlations (Figure 4). Additionally, linear models exhibited more significant differences throughout the stance phase than the recurrent neural network, when comparing subject−average curves (Figure 5). 

The linear models exhibited a systematic (or significant) increase in the vertical GRF in late stance (*p* < 0.05, generally between 75 and 100% stance, Figure 5). The average error in this region across all subjects and conditions was 5.6% for the linear model and 4.0% for the recurrent neural network. In the first 20% of stance, both models exhibited higher errors in the vertical GRF (linear models: 6.7%, recurrent neural network: 5.5%), but they were not significantly different. 

For the A/P GRF, the linear model exhibited significant differences in both early (*p* < 0.05, generally between 20 and 30% of stance) and late stance (*p* < 0.05, 70–100% of stance Figure 6, middle row) as this model was unable to predict peak breaking nor peak propulsive A/P GRFs. The average error across all subjects and conditions in the linear model between 20% and 30% of stance was 6.7% and the recurrent neural network was 5.0%. This region also represented where there were some of the greatest errors in the linear model for A/P GRF prediction. The average error across all subjects and conditions in the linear model between 70 and 100% of stance was 5.8% and the recurrent neural network was 4.2%. The recurrent neural network had the highest errors (though non−significant, *p* > 0.05) between 0 and 20% of stance where the average error was 5.5%.

### 3.2. Model Ability to Predict Step−Average Ground Reaction Forces

Across all speeds and slopes, both the recurrent neural network and the linear models exhibited significant differences between the predicted and true vertical and A/P GRFs for individual subjects (*p* < 0.05, see Figure 6 for recurrent neural network performance). These areas of significant difference varied from subject to subject (Figure 6); however, generally, there were significant differences at the peak vertical GRF and the peak breaking and propulsion peaks in the A/P GRF. The peak vertical GRF also corresponded to the largest percent error across all subjects. The largest error in the A/P GRF occurred in the first 20% of stance for the recurrent neural network (Figure 6).

### 3.3. Effect of Speed and Slope

Removing speed and slope as predictors from the models had very little effect on RMSE and correlation (Figure 4) between the models and the true GRFs. Removing these two predictors did not increase the number of significant regions for the recurrent neural network nor for the linear models. 

## 4. Discussion

This was the first running study to predict both the vertical and A/P GRFs during running over different speeds and slopes. Such predictions can be harnessed to better understand lower limb biomechanics outside of the laboratory. For instance, these wearable−based GRFs could compute tibia bone loads [1] to inform athletes of potential overuse injuries. Similarly, these GRFs could be paired with other wearables, such as inertial measurement units that estimate joint angles [3], to estimate lower limb joint powers. These joint powers could provide runners with real−time performance metrics or allow running researchers to understand the effects of high−performance footwear in realistic running conditions. We also introduced a new combination of tools to understand the performance of model predictions: statistical parametric mapping (SPM) combined with time−continuous error. The SPM provided information about where in stance the model prediction was significantly different from the measured GRF. For instance, the linear models exhibited a systematic bias in the vertical GRF in late stance and near the breaking and propulsive A/P GRF peaks (Figure 5). Such a statistical assessment should be accompanied with a time−continuous error measurement as SPM does not provide an indication of *HOW* different; rather there is a region *THAT* is different.

Removing speed and slope as predictors from both model types had little impact on their respective performances (Figure 4). In future wearable applications this means that the GRF prediction errors shown here will not inflate due to further estimations of running speed or running slope. With IMUs attached to the foot, running speed can be estimated within 5% of the true value [30]; however, it is currently an open question whether ground slope can be estimated with IMUs during running. The fact that these two predictors have little impact on accuracy could be interpreted as plantar pressure contains a rich enough dataset to discriminate between running on different slopes and speeds. Removing speed as a predictor most likely did not affect the GRF estimates as it is well known that GRFs (and thus plantar pressure) increase in magnitude with speed [31]. On the other hand, there is no known relationship between running slope and plantar pressure. One possibility is that the running slope could change the foot contact patterns resulting in different time−continuous plantar pressures. However, visual examination of the vertical GRFs for all subjects and trials showed that 11 subjects changed to a forefoot striking pattern with uphill running, two subjects changed to a rearfoot striking pattern with downhill running and five subjects did not change foot contact pattern for the different slopes. As these contact patterns did not consistently change, there likely is a more complex relationship between plantar pressure and running slope. Other models that predicted running vertical GRFs on slopes utilized IMU signals as model inputs along with running speed and slope [12]. It is currently not known if these IMU−based models are sensitive when removing speed and slope as predictors.

It is not surprising that the recurrent neural network outperformed the linear models in predicting vertical and A/P GRFs (Figure 4). Previous studies that compared linear models vs. machine learning algorithms have also shown this discrepancy [10,32]. The linear models did show a similar correlation between the prediction and the vertical GRF as the recurrent neural network. This could be due to the fact that the force normal to the foot surface (i.e., what the plantar pressure measures) is almost entirely in the vertical direction. The advantages of the recurrent neural network can be observed in the predictions of the A/P GRF—where the correlation and RMSE were both better than the linear model (Figure 4). Such an improvement in performance may be partially attributed to how the recurrent neural network predicts each time point in the GRF curve, whereas the linear models provide a scaling factor for the entire stance phase for each of the five plantar pressure curves (Figure 2). This increased prediction power of the recurrent neural network especially aided in estimating A/P GRFs for forefoot striking runners. For instance, one subject that was a forefoot striker for all conditions had an average correlation of 0.83 and RSME of 0.12 BW with the linear models and an average correlation of 0.96 and average RMSE of 0.06 BW. For this subject, the linear models anticipated pressure in the heel region in early stance. As this pressure was not present, the linear models’ predictions suffered, yet the recurrent neural network did not have trouble predicting this subject’s A/P GRF. In the future, methodologies similar to layer−wise relevance propagation [33,34] could be adapted for recurrent neural networks in order to understand the impacts of the plantar pressure during certain times in the stance phase of GRF prediction. 

One noticeable difference in the recurrent neural network performance was how the model performed on average (Figure 5) versus for individual subjects (Figure 6). For individual subjects, there were times in the step where there were systematic differences (i.e., significant differences) between the model and the ground truth; whereas, on average, the recurrent neural network exhibited no systematic differences. This discrepancy can be linked to how the recurrent neural network is trained. Here we performed a leave−one−subject−out validation, which means the model optimized the GRF prediction and mitigated its error for 17 subjects across all of their steps. Then, this model is applied to a single subject that has relatively low variability (with respect to 17 subjects). This will inevitably lead to systematic differences between the GRF prediction and the ground truth. Similar effects have also been observed in GRF predictions from other recurrent neural networks [12]. As currently trained, the recurrent neural network performs very well overall; but it would be difficult to have confidence in secondary biomechanical metrics (e.g., tibia bone load) computed from these predictions for individual subjects. 

In order to improve the recurrent neural network’s performance for individual subjects, subject−specific training can be used to enhance GRF estimates. For instance, Figure 7 shows the difference that this subject−specific training can have on the recurrent neural network performance. Here the model was set up similarly as described in the method with some slight deviations. Seventeen subjects were used to create a “generic” model. Then a “subject−specific” model was created by training that “generic” model with ≈10% of the steps (165 steps) for the subject of interest. This scenario would mimic a calibration session where a new user briefly comes into the laboratory and runs several minutes on an instrumented treadmill. The subject illustrated in Figure 7 exhibited the greatest combined error for the vertical and A/P GRF prediction in order to demonstrate the maximum potential of this subject−specific training. For this example, the subject−specific model reduced the significantly different regions across all conditions by 77% in the vertical GRF and 53% in the A/P GRF. This subject−specific model also reduced the average percent error across all conditions from 7.7% to 2.9% for the vertical GRF and 8.0% to 5.2% for the A/P GRF. Such a low error rate could provide confidence to researchers when computing additional metrics from the GRFs, such as tibia bone load [1]. Additionally, such models that predict accurate GRFs could be used in conjunction with IMUs that provide joint kinematics [3] to understand how high−performance footwear (e.g., Nike VaporFly) affects lower limb kinetics outside of the lab.

There are several limitations to acknowledge with this study. This study was performed exclusively on an instrumented treadmill and the speed provided to the recurrent neural network was the speed set for the treadmill. Future studies should also examine the affects of recurrent neural network predictions overground and with different surfaces, which can change the GRFs [35]. Here we explored a wide range of running slopes; however, it is not known if even greater slopes would decrease the accuracy of a recurrent neural network that does not utilize slope as an input. The subjects recruited were young, healthy, and active. It is not known how the models shown here would perform with elderly or clinical population, which can have different time−continuous ground reaction forces [36,37]. 

### Anecdotes from Model Building

In this sub−section, we wanted to explain and/or elucidate decisions that were made in the creation of the GRF prediction model in order to aid future researchers who are interested in the outlined methodologies. 

The five plantar pressure regions examined here were based on data exploration and preliminary linear model fitting. We explored as little as three regions and some explorations looked at regions that were unequal in size/length. The five regions used here worked relatively well. We did explore using all 99 pressure sensors as inputs to the recurrent neural network; however, it did not improve performance enough to justify the added complexity and reduced applicability to other pressure sensing modalities.The GRFs were aligned parallel/perpendicular to the gravity vector as preliminary exploration demonstrated that such an orientation enabled better linear model predictions in contrast to GRFs aligned parallel/perpendicular to the running surface.Including a binary predictor variable for left/right foot was explored; however, it did not affect model performanceThe recurrent neural network sequence input layer was responsible for normalizing the predictor variables. We found that ‘Z−Score’ normalization resulted in the best performance. During network development we also experimented with four other methods, which did not perform as well:○‘zerocenter’; Subtract the mean○‘Rescale−symmetric’; Rescale range to [−1 1]○‘Rescale−zero−one’; Rescale range to [0 1]○‘none’; Raw inputsWe applied a dropout function to each of the bidirectional LSTM layers of the recurrent neural network that set randomly selected nodes to 0. This was done to prevent overfitting and the dropout probability was 30% and 20% for the first and second bidirectional LSTM layer, respectively. During development, we experimented with lower (up to 0%) and higher (up to 80%) dropout probabilities. Generally, higher probabilities resulted in worse performances, while lower dropout probabilities created better training results but worse testing results.For the first two fully connected layers of the recurrent neural network we applied the hyperbolic tangent as activation function, while the last fully connected layer was not exposed to an additional transfer function. We experimented with the rectified liner unit transfer function as an alternative but found no substantial differences within the network performances.

## 5. Conclusions

Recurrent neural networks estimated average vertical and A/P GRFs more accurately than linear models across all speeds and slopes. This was particularly evident with the A/P GRF, as the recurrent neural network reduced the average root mean squared error by 55% with respect to the linear model. Subject−specific ground reaction force predictions were enhanced with subject−specific model training (i.e., including steps from the subject in the training data). The improvement for one subject reduced prediction errors from 7.7% to 2.9% for the vertical GRF and 8.0% to 5.2% for the A/P GRF. With such low errors, these wearable−based GRFs could be confidently used outside of the laboratory to understand running performance or injuries.

## 6. Patents

Work shown here is included in patent 63/291,424 and 63/315,847.

## Figures and Tables

**Figure 1 sensors-22-03338-f001:**
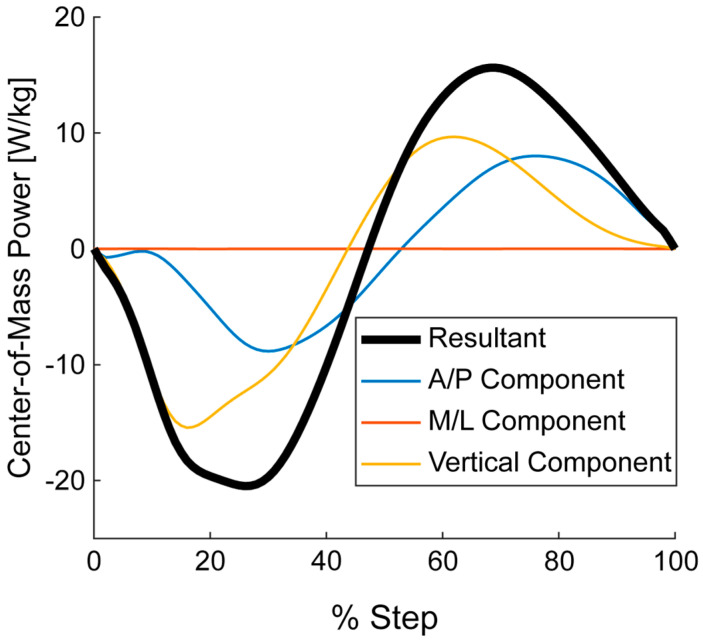
Center−of−Mass (COM) power components for running at 3.0 m/s on level ground. The three different components of the COM power were computed respectively from the anterior/posterior (A/P), medial/lateral (M/L), and vertical components of the ground reaction force. The sagittal plane components (i.e., vertical and A/P) have the greatest contribution to the COM power. Presented data are from one representative subject.

**Figure 2 sensors-22-03338-f002:**
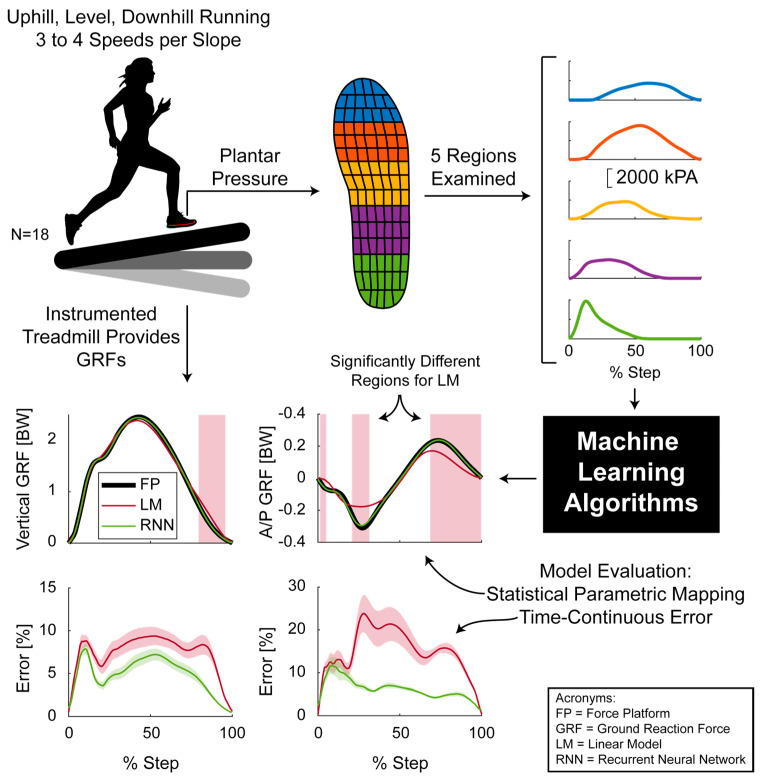
Vertical and anterior−posterior (A/P) ground reaction force prediction using plantar pressure. The plantar pressure was segmented into five different regions (indicated by the different colors). Example plantar pressure curves are shown in the same color to the right of the respective region. Additional inputs to the machine learning algorithms include running speed, running slope, and subject mass. Model performance was then evaluated using statistical parametric mapping, time continuous error (study mean ± standard deviation), root mean square error (not shown here), and correlation coefficients (not shown here).

**Figure 3 sensors-22-03338-f003:**
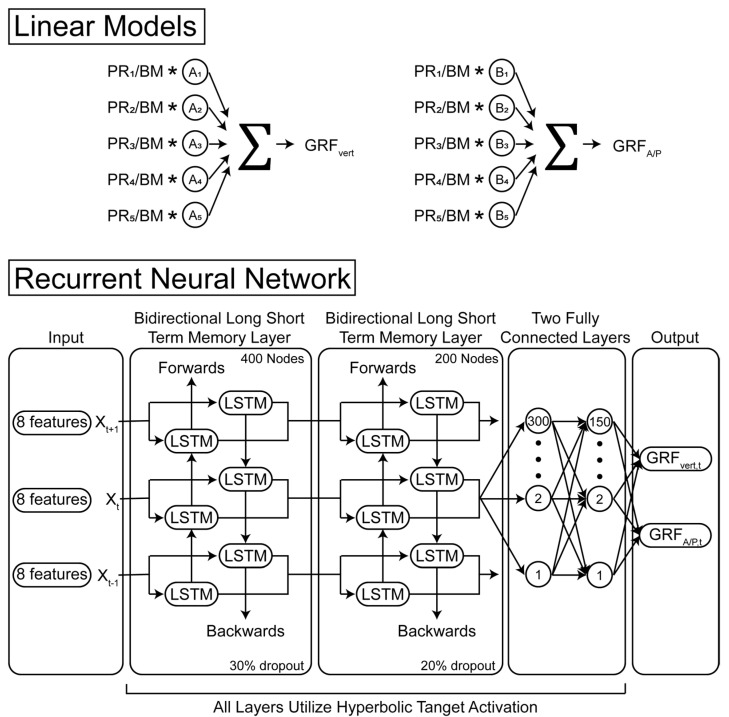
Linear and recurrent neural network model architecture for computing vertical (vert) and anterior–posterior (A/P) ground reaction forces (GRF). The linear models predicted vertical and A/P GRFs from five different pressure regions (PR_1–5_) divided by subject−specific body mass (BM). The coefficients (A_1–5_, B_1–5_) were computed using a least−square regression. The recurrent neural network predicted GRFs based on eight different inputs: PR_1–5_, body mass, speed and slope. The bidirectional long short−term memory (LSTM) layers utilized information from the current sample (x_t_) and from the future (x_t+1_) and previous samples (x_t-1_). For example, the GRF at 1% of the step is informed from the input at 0%, 1%, and 2% of the step. Both models were cross−validated using a leave−one subject out approach.

**Figure 4 sensors-22-03338-f004:**
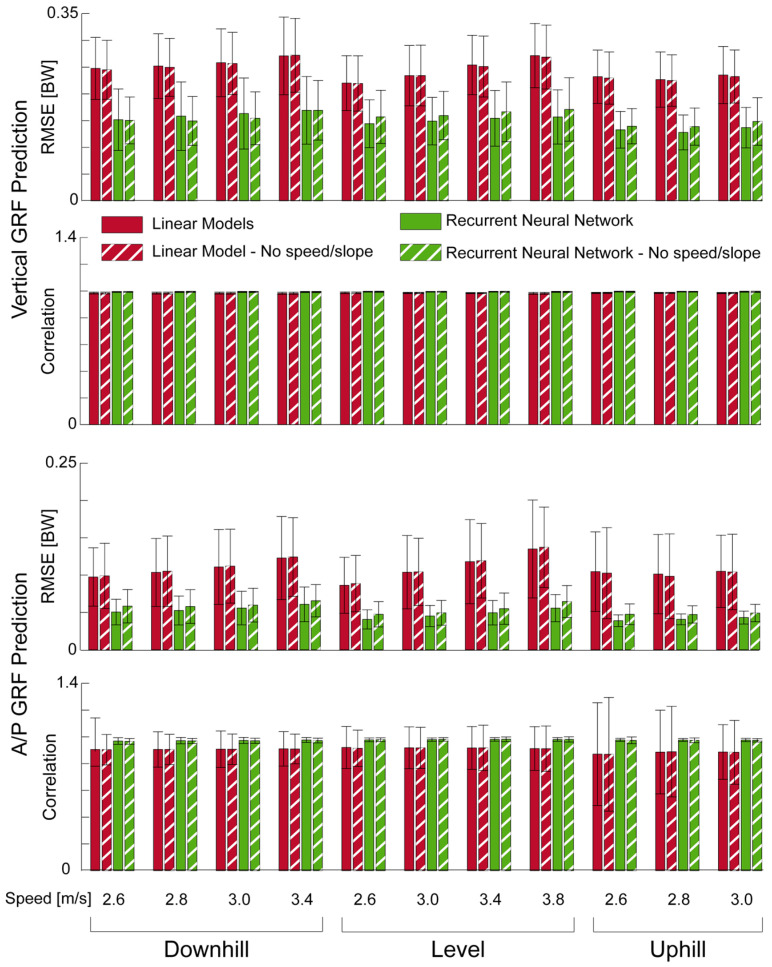
Average (N = 18) summary metrics for vertical and anterior/posterior (A/P) ground reaction force (GRF) prediction. Downhill and uphill running was performed at 6° (10.5% grade). Error bars represent standard deviations.

**Figure 5 sensors-22-03338-f005:**
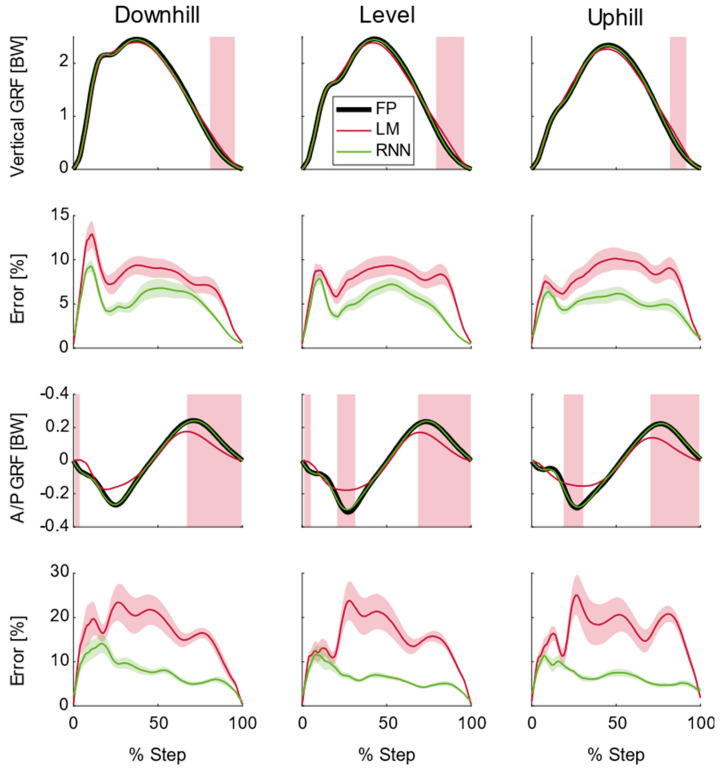
Average (N = 18) ground reaction forces (GRFs), model predictions, and model errors for running at 3.0 m/s. Top row shows average vertical GRF from the force platform (FP) along with average linear model (LM) and recurrent neural network (RNN) predictions. Areas shaded red are where there were significant differences between the FP and LM (*p* < 0.05). There were no significant differences between the FP and the RNN (*p* > 0.05). The second row shows the mean absolute percent error plus/minus one standard deviation for each model in the vertical direction. The third row shows average anterior/posterior (A/P) GRF from the FP along with LM and RNN predictions. Areas shaded red are where there were significant differences between the FP and LM (*p* < 0.05). There were no significant differences between the FP and the RNN (*p* > 0.05). The last row shows the mean absolute percent error plus/minus one standard deviation for each model in the A/P direction. Downhill and uphill running was performed at 6° (10.5% grade).

**Figure 6 sensors-22-03338-f006:**
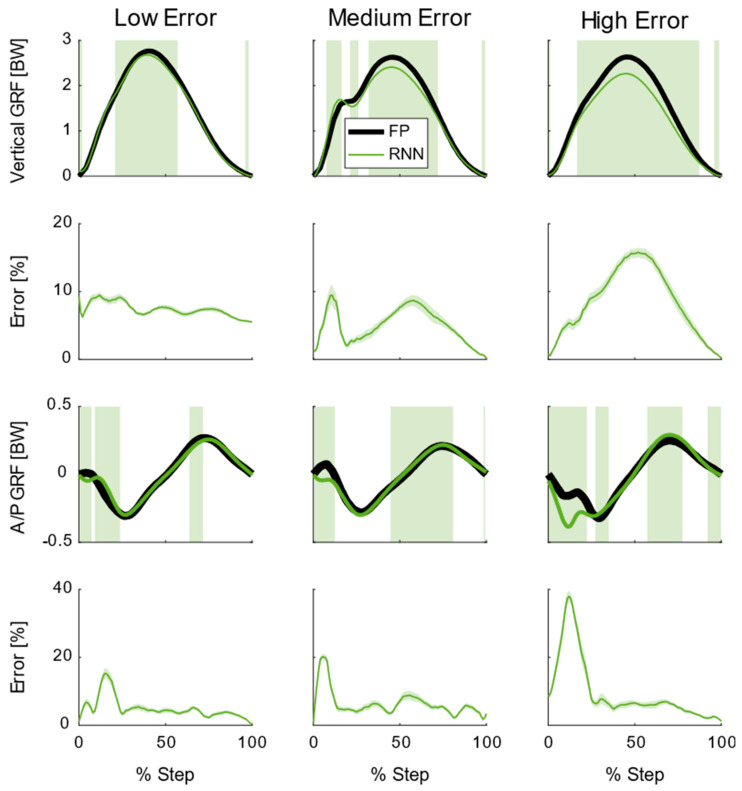
Subject−averaged (N = 18) ground reaction forces (GRFs), model predictions, and model errors for running at 3.0 m/s on level ground. The top row shows average vertical GRF from the force platform (FP) along with average recurrent neural network (RNN) predictions. Each column represents a different subject that either had the lowest, near average, and highest average errors in the RNN prediction of vertical GRF. Areas shaded green are where there were significant differences between the FP and RNN (*p* < 0.05). The second row shows the mean absolute percent error plus/minus one standard deviation for the RNN in the vertical direction. The third row shows average anterior/posterior (A/P) GRF from the FP and average RNN predictions. Each column represents a different subject that either had the lowest, near average, and highest average errors in the RNN prediction of A/P GRF. Areas shaded green are where there were significant differences between the FP and RNN (*p* < 0.05). The bottom row shows the mean absolute percent error plus/minus one standard deviation for the RNN in the A/P direction.

**Figure 7 sensors-22-03338-f007:**
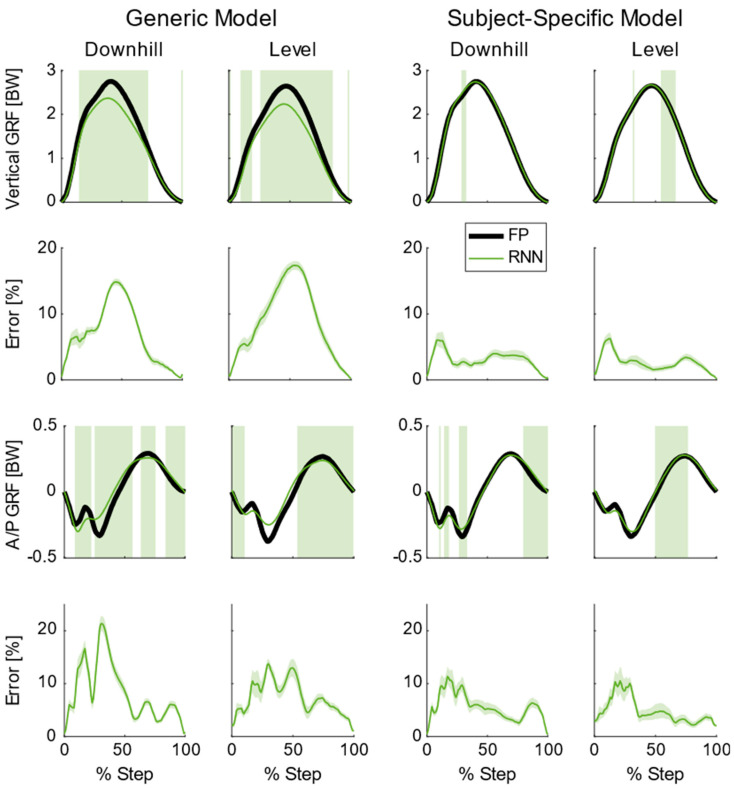
Subject−averaged (N = 18) ground reaction forces (GRFs), model predictions, and model errors for running downhill and level ground at 3.4 m/s for a single subject. The left two columns represent a generic model that was trained on the other 17 subjects in the study. The right two columns represent a subject−specific model that is trained on 17 subjects plus ≈10% of the data from the subject illustrated. The top row shows the average vertical GRF from the force platform (FP) along with average recurrent neural network (RNN) predictions. Areas shaded green are where there were significant differences between the FP and RNN (*p* < 0.05). The second row shows the mean absolute percent error plus/minus one standard deviation for the RNNs in the vertical direction. The third row shows average anterior/posterior (A/P) GRF from the FP and average RNN predictions. Areas shaded green are where there were significant differences between the FP and RNN (*p* < 0.05). The bottom row shows the mean absolute percent error plus/minus one standard deviation for the RNNs in the A/P direction.

## Data Availability

Data is available at 10.5281/zenodo.6485220.

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
