# Peer review of "Estimating Running Ground Reaction Forces from Plantar Pressure during Graded Running"

_sensors, 2022, doi:10.3390/s22093338_

Round 1

Reviewer 1 Report

The manuscript focuses on the ratings of running GRFs outside the lab. It provides interesting and biomechanically relevant facts about the essentials of GRF assessment. 18 participants ran on an instrumented treadmill and GRFs and Pedar plantar pressure from both the left and right foot were collected.

One of the elements of originality is prediction not only of vertical but also anterior-posterior GRFs across different speeds and slopes. Linear and RNN models were developed to predict these GRFs from plantar pressure data. Another interesting aspect is that GRF predictions were compared with the ground truth using statistical parametric mapping that allowed the determination of time-continuous accuracy. The article presents very interesting and significant results for future researchers in the field of GRF analysis. Looking into the presented results, researchers can use a new set of tools with wearable-based GRFs to understand running performance or injuries in real-world settings. Assumptions and their confirmations about the possibilities of using prognostic tools for GRF assessment outside the lab are also provided.

The paper is easy to read, interesting and provides interesting scientific insights and interpretations. 4.1. Anecdotes from model building - very original, but at the same time a very valuable subdivision. Maybe it could be mandatory!! I really liked it - thank you. I think the investigation needs to be continued. However, I also have a few minor remarks.

Minor comments:

  • The summary does not represent the whole work clearly enough. Since the purpose of the publication is to show how accurately GRFs can be predicted using plantar pressure data, more should be said about those tools: subject-specific and so…
  • 18 subjects participated in the research. Has a sample been calculated to confirm the results? Is it enough? What data sample was used?
  • The methodological part is very clear, but a study diagram showing the order, sample size, and other conditions would certainly help.
  • Measurements were performed on both feet. How were the results of both legs evaluated? Separately or together? Why?
  • Since the results are really many, I highly suggest the author think carefully and present the final key conclusions in place of a summary. It now appears that the authors are very careful in presenting the results obtained. Is it true?

Author Response

Responses to the reviewer are in green. (please see attached document for colors)

The manuscript focuses on the ratings of running GRFs outside the lab. It provides interesting and biomechanically relevant facts about the essentials of GRF assessment. 18 participants ran on an instrumented treadmill and GRFs and Pedar plantar pressure from both the left and right foot were collected.

One of the elements of originality is prediction not only of vertical but also anterior-posterior GRFs across different speeds and slopes. Linear and RNN models were developed to predict these GRFs from plantar pressure data. Another interesting aspect is that GRF predictions were compared with the ground truth using statistical parametric mapping that allowed the determination of time-continuous accuracy. The article presents very interesting and significant results for future researchers in the field of GRF analysis. Looking into the presented results, researchers can use a new set of tools with wearable-based GRFs to understand running performance or injuries in real-world settings. Assumptions and their confirmations about the possibilities of using prognostic tools for GRF assessment outside the lab are also provided.

The paper is easy to read, interesting and provides interesting scientific insights and interpretations. 4.1. Anecdotes from model building - very original, but at the same time a very valuable subdivision. Maybe it could be mandatory!! I really liked it - thank you. I think the investigation needs to be continued. However, I also have a few minor remarks.

I would like to first thank the reviewer for their time, review, and kind words. The section pointed out (4.1 Anecdotes from model building) was one that we felt could reduce the barrier to entry for future researchers who wish to enter into this field.

Minor comments:

The summary does not represent the whole work clearly enough. Since the purpose of the publication is to show how accurately GRFs can be predicted using plantar pressure data, more should be said about those tools: subject-specific and so…

We have updated our introduction to include the contributions (in bulleted form) of the manuscript, in place of the summary at the end of the article:

“The main contributions of this work are:

  • The first running study to predict multiple ground reaction force components during running for different speeds and slopes
  • We introduced a new combination of tools to understand the performance of time-continuous model predictions during gait
  • GRF predictions with plantar pressure do not need a priori knowledge of the speed or slope
  • Subject-specific training can enhance GRF predictions, such that these predictions could be confidently used outside of the lab.”

 We have also changed the “Summary” section to a “Conclusions” section (as suggested in the next comment) and updated the text to the following:

“Recurrent neural networks estimated average vertical and A/P GRFs more accurately than linear models across all speeds and slopes. This was particularly evident with the A/P GRF as the recurrent neural network reduced the average root mean squared error by 55% with respect to the linear model. Subject-specific ground reaction force predictions were enhanced with subject-specific model training (i.e. including steps from the subject in the training data). This improvement for one subject reduced prediction errors from 7.7% to 2.9% for the vertical GRF and 8.0% to 5.2% for the A/P GRF. With such low errors, these wearable-based GRFs could be used outside of the lab confidently to understand running performance or injuries.”

Since the results are really many, I highly suggest the author think carefully and present the final key conclusions in place of a summary. It now appears that the authors are very careful in presenting the results obtained. Is it true?

As suggested, we have replaced the summary with the key conclusions that relate to the purpose statement. For the new conclusion please see the response to the previous comment.

18 subjects participated in the research. Has a sample been calculated to confirm the results? Is it enough? What data sample was used?

We examined a post-hoc power analysis for correlations (using the R package pwr). All of the results showed a 100% power as the correlations for both the recurrent neural network and the linear model were above 0.90. We did not perform an a-priori power analysis for this study as there is not a power analysis for regression-style machine learning algorithms (machine learning power analyses have been published for classification algorithms). If the results in our manuscript were to be used for a correlation power analysis with 80% power, it would require 5 subjects. In our opinion, five subjects would not provide diverse biomechanical data to adequately examine the recurrent neural network evaluated here. Our recurrent neural network was based off of previous work (Alcantara et al. 2022), which created and tested a recurrent neural network with a similar number of subjects. We have included the following text to reflect this:

“A similar study size was previously utilized to examine the performance of machine learning algorithms predicting GRFs [12]”

The methodological part is very clear, but a study diagram showing the order, sample size, and other conditions would certainly help.

We have updated our methodology figure (Figure 2) to illustrate the study conditions, sample size, and order of operations performed in this study.

Measurements were performed on both feet. How were the results of both legs evaluated? Separately or together? Why?

We have updated our “Validation” portion of the manuscript to include:

Left and right foot GRFs were evaluated together.”

And in the discussion, section 4.1 we included the following bullet:

  • “Including a binary predictor variable for left/right foot was explored; however, it did not affect model performance”

Reviewer 2 Report

This study aimed to Estimate Running Ground Reaction Forces from Plantar Pressure during Graded Running.  I have the following major suggestions.

  1. What is the novelty of this study although several Estimating Running Ground Reaction Forces from Plantar Pressure during Graded Runningapproaches have been studied earlier?
  2. Please add a paragraph about the contribution of this article in a bulleted form at the end part of the Introduction section.
  3. Authors should review gait changes due to diseases and improve references mentioning studies of various neuromuscular changes in article, prediction of myoelectric biomarkers in post-stroke gait.
  4. Authors should add a figure of the experimental protocol used in this study.
  5. Authors should report the details of the dataset used in prediction modeling. How did the authors deal with dataset imbalance challenges?
  6. Both training and testing ROC curves need to be shown.
  7. Authors need to mention the model parameters or hyperparameters.
  8. Authors should present the training and validation accuracy graphs of the proposed model with changes in the size of the dataset.
  9. Authors should review gait changes due to stroke and improve references mentioning studies of various neuromuscular changes in article, real-time gait monitoring system for consumer stroke prediction service.
  10. Authors should provide an error-bar plot (also * mark if significant) of all main results stated in Tables for better visualization.
  11. Authors should report more performance measures of classifiers, such as accuracy, sensitivity, specificity, and precision from the prediction model.
  12. Authors should discuss the strength and weaknesses of reported findings with other previous findings and improve manuscript references, as suggested, in the discussion section.

Author Response

This study aimed to Estimate Running Ground Reaction Forces from Plantar Pressure during Graded Running.  I have the following major suggestions.

Thank you for your time and your comments. Responses to the reviewer are in green. (please see attached version for colors)

  1. What is the novelty of this study although several Estimating Running Ground Reaction Forces from Plantar Pressure during Graded Running approaches have been studied earlier?

We performed a ground reaction force (GRF) prediction literature review in PubMed that examined both pressure and IMU based estimates. From this literature review, we found 29 relevant articles. 12 of these articles examined running and one of the 12 articles estimated GRFs from custom-built pressure sensors. Additionally, only one of the 12 articles estimated GRFs (from IMUs) on different running grades (Alcantara et al. 2022). There were two articles that utilized plantar pressure (Fukushi et al. 2019, Wei et al. 2019) to estimate multiple ground reaction forces during walking. We have ensured that these three specific papers appear in our References as follows:

“8. Wei, F.; Crechiolo, A.; Haut, R.C. Prediction of Ground Reaction Forces in Level and Incline/Decline Walking from a Multistage Analysis of Plantar Pressure Data. J. Biomech. 2019, 84, 46–51, doi:10.1016/j.jbiomech.2018.12.015.

  1. Fukushi, K.; Sekiguchi, Y.; Honda, K.; Yaguchi, H.; Izumi, S.-I. Three-Dimensional GRF and CoP Estimation during Stair and Slope Ascent/Descent with Wearable IMUs and Foot Pressure Sensors. In Proceedings of the 2019 41st Annual International Conference of the IEEE Engineering in Medicine and Biology Society (EMBC); July 2019; pp. 6401–6404.
  2. Alcantara, R.S.; Edwards, W.B.; Millet, G.Y.; Grabowski, A.M. Predicting Continuous Ground Reaction Forces from Accelerometers during Uphill and Downhill Running: A Recurrent Neural Network Solution. PeerJ 2022, 10, e12752, doi:10.7717/peerj.12752.”

In order to convey the novelty of the study we have also included a bulleted list (as suggested in comment #2) of the main contribution of this manuscript:

“The main contributions of this work are:

  • The first running study to predict multiple ground reaction force components during running for different speeds and slopes
  • We introduced a new combination of tools to understand the performance of time-continuous model predictions during gait
  • GRF predictions with plantar pressure do not need a priori knowledge of the speed or slope
  • Subject-specific training can enhance GRF predictions, such that these predictions could be confidently used outside of the lab.”

  1. Please add a paragraph about the contribution of this article in a bulleted form at the end part of the Introduction section.

We have included the contributions as suggested. For full text please see our response to comment #1.

  1. Authors should review gait changes due to diseases and improve references mentioning studies of various neuromuscular changes in article, prediction of myoelectric biomarkers in post-stroke gait.

While we appreciate the scientific contributions provided by Hussain and Park, “Prediction of Myelectric Biomarkers in Post-Stroke Gait”, we do not feel that it is appropriate to cite this article in our manuscript that predicts ground reaction force from plantar pressure. We have included as a limitation within our study it is unknown how well our results apply to clinical populations – specifically citing changes in post-stroke gait:

“The subjects recruited were young, healthy, and active. It is not known how the models shown here would perform with elderly or clinical population, which can have different time-continuous ground reaction forces [35,36].”

  1. Franz, J.R.; Kram, R. Advanced Age Affects the Individual Leg Mechanics of Level, Uphill, and Downhill Walking. J. Biomech. 2013, 46, 535–540, doi:10.1016/j.jbiomech.2012.09.032.
  2. Sharma, S.; McMorland, A.J.C.; Stinear, J.W. Stance Limb Ground Reaction Forces in High Functioning Stroke and Healthy Subjects during Gait Initiation. Clin. Biomech. 2015, 30, 689–695, doi:10.1016/j.clinbiomech.2015.05.004.

  1. Authors should add a figure of the experimental protocol used in this study.

We have updated Figure 2 in the Methods portion of the manuscript to include more details of the experiment.

  1. Authors should report the details of the dataset used in prediction modeling. How did the authors deal with dataset imbalance challenges?

We included the following in our results:

“All subjects across all conditions contributed to the training and testing datasets. In total, 24882 steps were analyzed.”

As no data were missing from our collection we did not evaluate dataset imbalances.

  1. Both training and testing ROC curves need to be shown.

It is common to report receiver operating characteristic (ROC) curves for binary classification machine learning problems. Specifically, the ROC curve examines the true positive rate (TPR) against the false positive rate (FPR). With this in mind, it is a valuable tool to scrutinize the performance of classification models. In the case of our manuscript, we utilize regression style models. Meaning, that given a certain set of input variables, a prediction is made on the corresponding vertical and anterior-posterior ground reaction forces. This prediction does not resemble a traditional classification problem as no true class can be assigned to the resulting predictions. In our opinion, providing a ROC curve would not enhance the manuscript. Furthermore, given that no true (or false) positive outcome can be derived from a regression model, reporting such a curve would be misleading to the readers. 

  1. Authors need to mention the model parameters or hyperparameters.

Within the manuscript we state:

The model consisted of 11 individual layers: a sequence input layer that processed eight predictor variables per time point (five pressure regions, mass, speed, and slope), a bidirectional long short-term memory layer (BiLSTM) with 400 nodes, a 30% dropout layer, a BiLSTM with 200 nodes, a 20 % dropout layer, two fully connected layers (FC) with 300 and 150 nodes that passed their respective activations through to a hyperbolic tangent layer, and a finally FC with 2 nodes that passed its activations to a regression output layer.”

We have also included the additional text regarding more specific hyperparameters:

“For more information regarding network training options and hyperparameters see the supplementary data.”

Please note that we will include the supplementary data with the final submission.

  1. Authors should present the training and validation accuracy graphs of the proposed model with changes in the size of the dataset.

We did examine how changes in the training set affected model performance. For this analysis we evaluated GRF predictions by randomly selecting nine subjects to train the model and then testing on the nine subjects that were not in the training set. We show the results from the leave-one-subject-out (LOSO) approach (data from the new Fig. 3) and the subject-specific models (LOSO+10% of subject data). All of this data is compiled in Table R3 on page 3 below. As more data is included in the model, the error is lower. We do not include this new data within the manuscript as the purpose of the manuscript was to: predict sagittal plane GRFs across different speeds and slopes, and to understand the time-continuous accuracy of the predictions on both a cohort and individual subject basis.”

As noted by Reviewer 1, we present many results in this manuscript, we feel that showing such an analysis would reduce the impact of the results already presented.

  1. Authors should review gait changes due to stroke and improve references mentioning studies of various neuromuscular changes in article, real-time gait monitoring system for consumer stroke prediction service.

While we appreciate the scientific contributions provided by Hussain, Park, et al. “Real-time Gait Monitoring System for Consumer Stroke Prediction Service”, we do not feel that it is appropriate to cite this article in our manuscript that predicts ground reaction forces from plantar pressure. We have included as a limitation within our study it is unknown how well our results apply to clinical populations – specifically citing changes in post-stroke gait. For manuscript text please see response #3.

  1. Authors should provide an error-bar plot (also * mark if significant) of all main results stated in Tables for better visualization.

We have removed Tables 1 and 2 and replaced it with a new Figure 3 as suggested.

  1. Authors should report more performance measures of classifiers, such as accuracy, sensitivity, specificity, and precision from the prediction model.

The linear model and recurrent neural network utilized here are regression style algorithms. It is common to evaluate classification with “accuracy, sensitivity, specificity, and precision”, as shown in Hussain and Park, “Prediction of Myelectric Biomarkers in Post-Stroke Gait” and supported by Huang and Ling “Using AUC and Accuracy in Evaluating Learning Algorithms”. It is common to present root mean squared error and correlation coefficients (or mathematical variations of these two outcome measures) for regression style machine learning algorithms (see Savelberg et al. 1999, Forner Cordero et al. 1999, Halilaj et al. 2018).

  1. Savelberg, H.H.C.M.; Lange, A.L.H. de Assessment of the Horizontal, Fore-Aft Component of the Ground Reaction Force from Insole Pressure Patterns by Using Artificial Neural Networks. Clin. Biomech. 1999, 14, 585–592, doi:10.1016/S0268-0033(99)00036-4.
  2. Forner Cordero, A.; Koopman, H.J.F.M.; van der Helm, F.C.T. Use of Pressure Insoles to Calculate the Complete Ground Reaction Forces. J. Biomech. 2004, 37, 1427–1432, doi:10.1016/j.jbiomech.2003.12.016.
  3. Halilaj, E.; Rajagopal, A.; Fiterau, M.; Hicks, J.L.; Hastie, T.J.; Delp, S.L. Machine Learning in Human Movement Biomechanics: Best Practices, Common Pitfalls, and New Opportunities. J. Biomech. 2018, 81, 1–11, doi:10.1016/j.jbiomech.2018.09.009.

We have clarified in our methods the type of machine learning algorithm used in our paper:

“We first estimated GRFs using linear regression models to provide a baseline com-parison for the regression style machine learning model that is outlined below.”

And

“We then designed a single regression recurrent neural network (RNN) to predict vertical and fore-aft ground reaction forces.”

  1. Authors should discuss the strength and weaknesses of reported findings with other previous findings and improve manuscript references, as suggested, in the discussion section.

We have increased the connections with prior literature and updated the text with:

“Other models that predicted running vertical GRFs on slopes utilized IMU signals as model inputs along with running speed and slope [12]. It is currently not known if these IMU-based models are sensitive removing speed and slope as predictors.”

 And

“It is not surprising that the recurrent neural network outperformed the linear models in predicting vertical and A/P GRFs (Fig. 3). Previous studies that compared linear models vs. machine learning algorithms have also shown this discrepancy [10,32].”

and

“Similar [subject-specific] effects have also been observed in GRF predictions from other recurrent neural networks [12].”

Table R1.  Root mean squared error, in bodyweights, of vertical and anterior posterior (A/P) ground reactions forces predicted with varying levels of training data. 9T signifies the model was trained with nine subjects and then tested with the remaining nine subjects. 17T signifies the model was trained with 17 subjects and tested on one subject. 17T+ signifies the model was trained with 17 subjects plus ≈10% of the subject’s data that is the test subject. Presented are means of the same nine subjects.

GRF

Model

6° downhill

Level

6° Uphill

2.6 m/s

2.8 m/s

3.0 m/s

3.4 m/s

2.6 m/s

3.0 m/s

3.4 m/s

3.8 m/s

2.6 m/s

2.8 m/s

3.0 m/s

Vertical

9T

0.15

0.15

0.15

0.17

0.14

0.14

0.16

0.17

0.14

0.13

0.15

17T

0.14

0.15

0.15

0.16

0.13

0.14

0.15

0.16

0.12

0.12

0.14

17T+

0.10

0.11

0.12

0.13

0.10

0.10

0.11

0.12

0.09

0.09

0.10

A/P

9T

0.05

0.05

0.05

0.05

0.04

0.04

0.05

0.05

0.04

0.04

0.05

17T

0.05

0.05

0.05

0.05

0.04

0.04

0.04

0.05

0.04

0.04

0.04

17T+

0.04

0.04

0.04

0.05

0.03

0.04

0.04

0.04

0.03

0.04

0.04

Round 2

Reviewer 2 Report

Thanks for addressing a few comments. Still few comments are not addressed.

  • Authors should provide a complete mathematical expression to estimate running ground reaction forces from plantar pressure.
  • Authors must address comment#8 in the last review “Authors should present the training and validation accuracy graphs of the proposed model with changes in the size of the dataset”.

Author Response

Thanks for addressing a few comments. Still few comments are not addressed.

Thank you for taking time to review our manuscript. Please see our responses in green. (please see the attached version for colored text).

Authors should provide a complete mathematical expression to estimate running ground reaction forces from plantar pressure.

We have updated our manuscript to include the following text and equations:

“We created a set of least square linear models to predict the vertical (Eqn. 1, GRFvert) and anterior/posterior GRFs (Eqn. 2, GRFA/P) from five different plantar pressure regions (PR1-PR5, Eqns. 1 & 2, Fig. 2). This linear regression accounted speed and slope by solving for the regression constants A1-A5 and B1-B5 for each different condition. This effectively enabled speed and slope to be independent predictors. The summed pressures from the five regions of the Pedar insoles (PR1-PR5) were normalized by subject body mass. This analysis was performed with these five different pressure regions in order to make the re-sults presented here applicable to other plantar pressure sensors that may not have as fine of a spatial resolution.

GRFvert = PR1*A1+ PR2*A2+ PR3*A3+ PR4*A4+ PR5*A5        (1)

GRFA/P = PR1*B1+ PR2*B2+ PR3*B3+ PR4*B4+ PR5*B5        (2)

We next created a set of linear models to understand the effect of different running speeds and running slopes on this model type. In this case, we created two linear models to predict the vertical and anterior-posterior GRF, respectively, to predict all running conditions. In this case, the regression constants A1-A5 and B1-B5 (Eqn. 1 & 2) were each solved for once. Similar to the first set of linear models, only the mass normalized insole pressures from the five different pressure regions were utilized to train the linear model.”

Also, as there is not a set or defined mathematical expression for a recurrent neural network, we have included the following.

“We then designed a single regression recurrent neural network (RNN) to predict vertical and anterior-posterior ground reaction forces. Recurrent neural networks have been used previously to estimate vertical GRFs [12]. Using such an algorithm allows for a more flexible solution than shown in Eqns. 1 & 2 as the network is not restricted to one certain type of mathematical operation.”

Authors must address comment#8 in the last review “Authors should present the training and validation accuracy graphs of the proposed model with changes in the size of the dataset”.

We have included an appendix that graphically shows the average root-mean-squared-error with changes in number of subjects included in the training data. We have also included the following in the Results section of our manuscript:

“Model performances with changes in training dataset size can be seen in Appendix A.”

Round 3

Reviewer 2 Report

Thanks for addressing the manuscript. Few more concerns as follow

  • Authors should provide a figure of the proposed framework of the regression model and RNN-model with all hyperparameters used in this study.

Author Response

Thanks for addressing the manuscript. Few more concerns as follow

  • Thank you for taking time to review our manuscript again.

Authors should provide a figure of the proposed framework of the regression model and RNN-model with all hyperparameters used in this study.

  • We have included a new figure (Figure 3) that elucidates both the linear regression model and the recurrent neural network.